# Coanda Effect Displayed in a Giant Intracranial Aneurysm

**DOI:** 10.3390/brainsci14090897

**Published:** 2024-09-05

**Authors:** Corneliu Toader, Petrinel Mugurel Rădoi, Ghaith Saleh R. Aljboor, Luca-Andrei Glavan, Razvan-Adrian Covache-Busuioc, Milena-Monica Ilie, Alexandru Vlad Ciurea

**Affiliations:** 1Department of Neurosurgery, “Carol Davila” University of Medicine and Pharmacy, 050474 Bucharest, Romania; corneliu.toader@umfcd.ro (C.T.); ghaith-saleh-radi.aljboor@drd.umfcd.ro (G.S.R.A.); luca-andrei.glavan0720@stud.umfcd.ro (L.-A.G.); razvan-adrian.covache-busuioc0720@stud.umfcd.ro (R.-A.C.-B.); milena-monica.ilie0720@stud.umfcd.ro (M.-M.I.); prof.avciurea@gmail.com (A.V.C.); 2Department of Vascular Neurosurgery, National Institute of Neurology and Neurovascular Diseases, 077160 Bucharest, Romania; 3Neurosurgery Department, Sanador Clinical Hospital, 010991 Bucharest, Romania

**Keywords:** Coanda effect, intracranial aneurysm, giant aneurysm, hemodynamics

## Abstract

The Coanda effect is a fluid dynamics phenomenon in which a fluid jet adheres to a convex or flat surface. This effect occurs when a liquid or gas jet emerging from an orifice clings to an adjacent surface and entrains the surrounding fluid, creating a lower-pressure region along its path that maintains its attachment to the surface. The Coanda effect accounts for the behavior of blood flow in the fetal right atrium and the dispersion of eccentric mitral regurgitation jets along atrial walls. This series of interesting images depicting a large 4 × 3.75 cm saccular intracranial aneurysm suggests that the Coanda effect may play a role in the pathophysiology of intracranial aneurysms and could be an underlying factor in their formation, progression, or rupture.

A 37-year-old female patient was admitted to our clinic with a history of mild left-sided hemiparesis and moderate cognitive impairments, which progressively developed during the postpartum period, as the patient gave birth six weeks previously.

A pterional craniotomy on the left side was performed. The dura mater, under marked tension, was circumferentially suspended and incised with a basal pedicle. Using the operating microscope, the carotid cistern on the left side was approached and the proximal Sylvian fissure was dissected. A giant, unruptured aneurysm (Figure 1) at the bifurcation of the internal carotid artery, measuring approximately 4 × 3.75 cm, was identified, with the M1 and both M2 (Figure 2 and Figure 3) segments of the middle cerebral artery originating from the aneurysmal sac (Figure 4). Due to the involvement of these arterial segments, direct clipping at the neck of the aneurysm was deemed impossible. If no significant arterial segment originated from the aneurysmal sac, it would have been feasible to clip the aneurysm, given its 2.5 mm neck. Alternatively, a bypass procedure could also have been achieved. Hemostatic material (Surgicel) was applied for local wrapping of the aneurysmal sac. Moderate cerebral collapse was noted. Hemostasis was achieved using electrocoagulation, Surgicel, and tamponade. The dura mater was sutured, and the bone flap was replaced over an epidural drain exiting through a burr hole. The wound was closed in anatomical layers and dressed.

The postoperative period was favorable, with significant neurological improvement noted.

## Figures and Tables

**Figure 1 brainsci-14-00897-f001:**
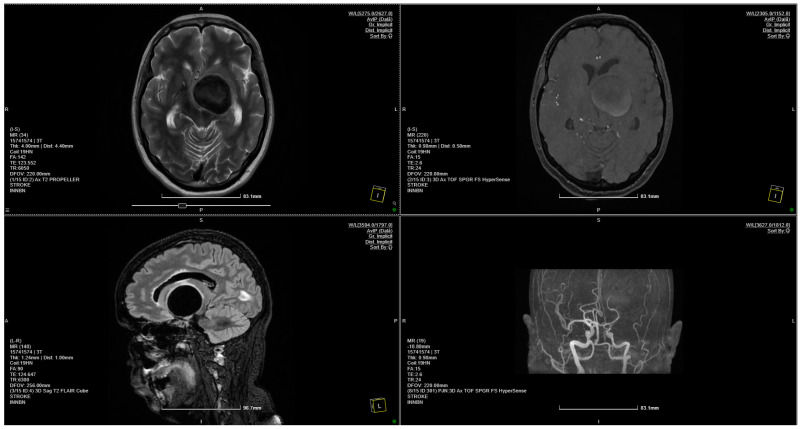
MRI scan showing a 4 × 3.75 cm giant aneurysm.

**Figure 2 brainsci-14-00897-f002:**
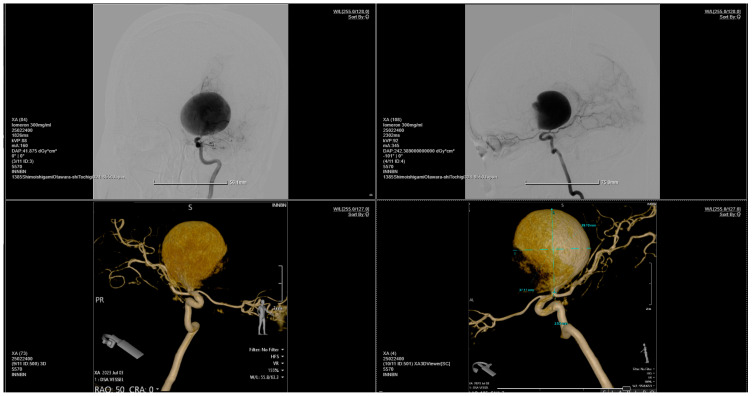
Bilateral carotid and vertebral angiography reveals a giant aneurysm located at the bifurcation of the right internal carotid artery, encompassing the M1 segment of the right middle cerebral artery. The right middle cerebral artery is not injected. Otherwise, the main arterial vessels of the brain appear normal.

**Figure 3 brainsci-14-00897-f003:**
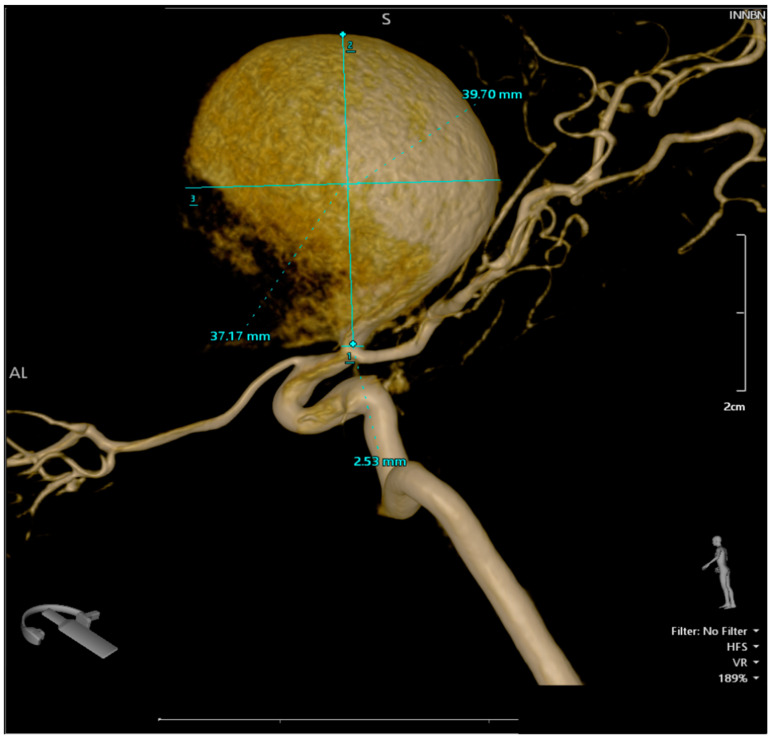
The 3D reconstruction of the rotational DSA illustrating a giant aneurysm with dimensions of 4 × 3.75 cm, located at the bifurcation of the right internal carotid artery, encompassing the M1 segment of the right middle cerebral artery.

**Figure 4 brainsci-14-00897-f004:**
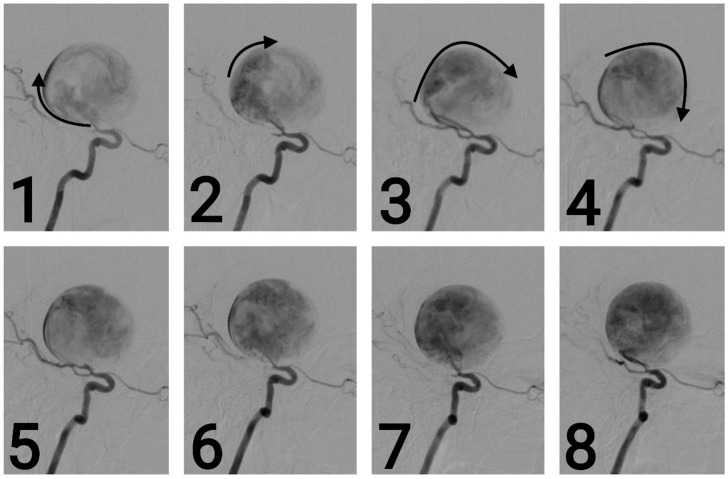
Series of 2D digital subtraction angiographies that highlight the Coanda effect [1]. The contrast substance tends to adhere to the walls when first entering the aneurysm—as the arrows show—and continues to follow the walls, as seen in the series of chronological images—from 1 to 8. [2]. This phenomenon also creates small vortexes inside the aneurysm, vortexes which are common findings during regurgitative valvopathies in cardiology [3].

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
