# Peer review of "Coanda Effect Displayed in a Giant Intracranial Aneurysm"

_brainsci, 2024, doi:10.3390/brainsci14090897_

Round 1

Reviewer 1 Report

Comments and Suggestions for Authors

The manuscript showed  an interesting conda effect in a large aneurysm. The manuscript could be accepted after minor revision.

1. In line 37. "4*3,75cm",punctuation was used incorrectly; 

2. How to deal with the giant aneuysm, clip or by pass? the detail was not descripted in the manuscript.

Author Response

Thank you for your valuable feedback and for recognizing the significance of our findings related to the Coanda effect in a large aneurysm. We appreciate your suggestions for improvement and have addressed them as follows:
Punctuation Correction: We have corrected the punctuation issue in line 37. The dimensions of the aneurysm are now accurately presented as "4 x 3.75 cm."
Management of the Giant Aneurysm: We acknowledge the need for further detail regarding the management options for the giant aneurysm.

Reviewer 2 Report

Comments and Suggestions for Authors

Understanding how the Coanda effect influences hemodynamics could be invaluable in performing surgical aneurysm treatments and postoperative interventions. By integrating knowledge of the Coanda effect into clinical practice guidelines, healthcare providers can benefit from standardized protocols and recommendations for aneurysm management.

This article offers a fascinating exploration of imaging techniques related to the Coanda effect's influence on hemodynamics. The visual data presented provide valuable insights that could greatly improve the understanding and management of aneurysm treatments. Therefore, the article is recommended for acceptance.

Author Response

Thank you for your positive assessment of our article and for highlighting the importance of understanding the Coanda effect in the context of hemodynamics and aneurysm management. We are pleased to hear that you found the exploration of imaging techniques and the insights provided to be valuable.

Reviewer 3 Report

Comments and Suggestions for Authors since there is a novel manuscript type called Interesting Images, then I do not have any further comments and suggest to publish. Comments on the Quality of English Language

Moderate changes. 

Author Response

Thank you for your suggestion to publish our manuscript. We are delighted that you found our work suitable for this category of manuscript and appreciate your support. We believe that the visual data presented in our manuscript offer significant insights and contribute to the understanding of the Coanda effect in aneurysm management.